# Anti-Atopic Effect of Isatidis Folium Water Extract in TNF-α/IFN-γ-Induced HaCaT Cells and DNCB-Induced Atopic Dermatitis Mouse Model

**DOI:** 10.3390/molecules28093960

**Published:** 2023-05-08

**Authors:** Ga-Yul Min, Tae In Kim, Ji-Hye Kim, Won-Kyung Cho, Ju-Hye Yang, Jin Yeul Ma

**Affiliations:** Korean Medicine (KM) Application Center, Korea Institute of Oriental Medicine, 70 Cheomdan-ro, Dong-gu, Daegu 41062, Republic of Korea; gayul8955@kiom.re.kr (G.-Y.M.); tikim@kiom.re.kr (T.I.K.); jkim2903@kiom.re.kr (J.-H.K.); wkcho@kiom.re.kr (W.-K.C.)

**Keywords:** Isatidis folium, *Isatis tinctoria* L., atopic dermatitis, keratinocytes, anti-inflammation

## Abstract

Isatidis folium or *Isatis tinctoria* L. is a flowering plant of the Brassicaceae family, commonly known as woad, with an ancient and well-documented history as an indigo dye and medicinal plant. This study aimed to evaluate the anti-atopic dermatitis (AD) effects of Isatidis folium water extract (WIF) using a 2,4-dinitrochlorobenzene (DNCB)-induced AD-like mouse model and to investigate the underlying mechanism using tumor necrosis factor-α (TNF-α) and interferon-γ (IFN-γ)-activated HaCaT cells. Oral administration of WIF reduced spleen weight, decreased serum IgE and TNF-α levels, reduced epidermal and dermal thickness, and inhibited eosinophil and mast cell recruitment to the dermis compared to DNCB-induced control groups. Furthermore, oral WIF administration suppressed extracellular signal-regulated kinase and p38 mitogen-activated protein kinase protein expression levels, p65 translocation from the cytoplasm to the nucleus, and mRNA expression of TNF-α, IFN-γ, interleukin (IL)-6, and IL-13 in skin lesion tissues. In HaCaT cells, WIF suppressed the production of regulated upon activation, normal T cell expressed and secreted (RANTES), thymus and activation-regulated chemokine (TARC), macrophage-derived chemokine (MDC), MCP-1, and MIP-3a, which are inflammatory cytokines and chemokines related to AD, and inhibited the mRNA expression of RANTES, TARC, and MDC in TNF-α/IFN-γ-stimulated HaCaT cells. Overall, the results revealed that WIF ameliorated AD-like skin inflammation by suppressing proinflammatory cytokine and chemokine production via nuclear factor-κB pathway inhibition, suggesting WIF as a potential candidate for AD treatment.

## 1. Introduction

Atopic dermatitis (AD) is one of the most common chronic inflammatory diseases of the skin, characterized by symptoms such as severe itching, abnormal epidermal hyperplasia, erythema, and eczematous skin rash [1]. AD affects 3% of infants, 20% of children, and 3–10% of adults worldwide and up to 20% of children and 10% of adults in high-income countries [2]. Patients with AD experience various problems, such as irritation due to clothing, sleep disturbance, itching, medical expenses, and dietary restrictions, which not only considerably affects their quality of life but also their guardians [3,4].

AD is a multifactorial and complex disease characterized by skin barrier dysfunction and abnormal immune response. In addition, certain immunological aspects have been recognized as etiological factors in AD, including the role of the epidermal barrier and consequent aberrant cytokine expression. Allergic triggers (food allergies), contact allergens, irritants, microorganisms, and disrupted epidermal barriers stimulate epidermal keratinocytes to directly initiate an inflammatory response and produce various pro-inflammatory mediators, such as cytokines and chemokines [5]. Exposure to natural allergens, climatic factors, and possible air pollutants alongside a genetic component reportedly play a role in AD development [3,6]. Several current studies aim to determine the link between these factors and AD and the underlying mechanisms [7,8,9].

AD that begins in infancy causes comorbidities, such as allergic rhinitis and allergic asthma in childhood; this progression is known as the “atopic march”. Many studies have revealed why AD treatment should begin at an early stage. Multiple therapeutic agents are being developed for the early treatment of atopy, and numerous studies targeting various mechanisms are also in progress [10]. Particularly, certain cytokines and chemokines recently identified in AD inflammation, including regulated upon activation, normal T cell expressed and secreted (RANTES; CCL5), thymus and activation-regulated chemokine (TARC; CCL17), macrophage-derived chemokine (MDC), monocyte chemoattractant protein-1 (MCP-1; CCL2), and macrophage inflammatory protein-3α (MIP-3α; CCL20; LARC; Exodus-1; Scya20), are potential therapeutic targets for AD. TARC and MDC constitute sources of minimally invasive biomarkers and are detected in the blood as potential biomarkers with strong and significant correlations with clinical treatment response in AD [11,12,13].

Topical corticosteroids and calcineurin inhibitors comprise the first-line treatment for AD according to various international guidelines. Short-term daily use and long-term intermittent application of topical corticosteroids relieve symptoms and improve the quality of life of patients [1]. However, long-term use can cause various adverse effects, such as erythematous edema, which is characterized by erythema, scaling, and edema alongside a burning sensation [14]. When topical treatment fails, systemic immunosuppressive therapies, including cyclosporine, methotrexate, and azathioprine and phototherapy, are considered. However, there remains a risk of recurrence and side effects. Therefore, developing safe and effective alternative drugs is continuously required [15].

“Isatidis folium” is the term for the dried leaves of *Isatis tinctoria* L., a flowering plant that belongs to the Brassicaceae family [16], commonly known as woad. It is a herbaceous biennial or short-lived perennial species, considered native to Central Asia, as confirmed via genetic analysis. However, it is also reportedly native to southeastern Russia, southwest Asia, and possibly parts of southeastern Europe and considered a noxious weed in most of the western United States [17]. *I. tinctoria* is widely used for medicinal purposes in traditional Chinese medicine, recognized as a pharmacopeia plant in Europe since 2011, and currently utilized in the cosmetic industry [18,19]. The seed oil and leaves (powder/extract) of this plant are used as cosmetics for skin and hair conditioning owing to their skin-softening and moisturizing properties, and its roots (powder/extract) are used as cosmetic raw materials because of their astringent and skin-protecting properties [17].

Isatidis folium is known by different names, depending on the medicinal part of the plant. The leaves are known as “Da-Qing-Ye” in China and “Dae-Cheong-yeop” in Korea, and the roots are known as “Ban-Lan-Gen” in China and “Pan-Ram-geun” in Korea. Its aerial components can be processed into a dried powder via a traditional fermentation process (“Qing-Dai” or “Cheong-dae”) following dehydration for long-term preservation [20,21]. Isatidis folium is well known as a traditional herb medicine with various physiological activities, such as antiviral [22,23], antiendotoxin activity [24], and anti-inflammatory [25] activities. Furthermore, reportedly, it promotes procollagen synthesis and inhibits matrix metalloproteinases (MMP) expression via antiwrinkle activity [26]. However, its anti-AD efficacy has not been well researched. Therefore, this study aims to provide a theoretical basis for the use of Isatidis folium leaves in the treatment of atopic dermatitis.

## 2. Results

### 2.1. Effects of Isatidis Folium Water Extract (WIF) on AD-like Symptoms in Mice

The anti-AD effects of WIF were evaluated using a 2,4-dinitrochlorobenzene (DNCB)-induced AD-like mouse model. After 30 days of induction and oral administration (WIF: 100, 200 mpk, DEX: Dexamethasone 1 mpk, mpk: mg/kg), morphological changes were observed, and spleen weight and serum immunoglobulin E (IgE) and tumor necrosis factor-alpha (TNF-α) levels were measured. WIF treatment alleviated typical AD symptoms, such as surface erythema, edema, and eczematous skin lesions, compared with the control group that received no treatment (Figure 1A). The spleen mass and weight of the control group that received DNCB significantly increased compared with the normal group (Figure 1B,C). However, WIF-treated mice exhibited a slight but significant decrease in spleen weight compared with the control group. In WIF-administered mice, the serum IgE and TNF-α levels were significantly lower than those of the control group (Figure 1D,E).

### 2.2. Effects of WIF on Epidermal and Dermal Thicknesses in AD-like Mouse Model

To investigate the effects of WIF on the skin of the AD-like mouse model, the dermal and epidermal thicknesses of the skin of the mice were determined via hematoxylin and eosin (H&E) staining. The epidermis and dermis were abnormally thickened in the control group that received DNCB (Figure 2A). Conversely, in WIF-administered mice, abnormal thickening was significantly alleviated in the epidermis (200 mpk) and dermis (100 and 200 mpk; Figure 2B,C). In Figure 2A, boldfaced E and D represent the epidermis and dermis, respectively.

### 2.3. Effects of WIF on Immune Cell Infiltration in AD-like Mouse Model

Histopathological analysis using H&E and toluidine blue staining was performed to investigate how WIF improves AD-like skin symptoms in DNCB-induced AD-like mouse models. Eosinophils and mast cells in the dermis were significantly increased in the DNCB-administered control group compared with the normal group, whereas eosinophil (Figure 3A, yellow arrows) and mast cell (Figure 3B, red arrows) infiltration was significantly reduced in the group orally administered with WIF compared with the control group (Figure 3).

### 2.4. Effects of WIF on Inflammation-Related mRNA and Protein Expression in AD-like Mouse Model

This study evaluated the effects of WIF on inflammation-related mRNA and protein expression levels in the dorsal skin of a DNCB-induced AD-like mouse model. Phosphorylated extracellular signal-regulated kinases (ERK) and p38 mitogen-activated protein kinase (MAPK) proteins were present in the dorsal skin tissues of the control mice that received DNCB. However, their levels were significantly reduced in WIF-treated mice (Figure 4A,B). In addition, DNCB-induced cytoplasm-to-nucleus translocation of nuclear factor-κB (NF-κB) in the dorsal skin tissue of the AD-like mice was significantly inhibited in mice that received WIF (Figure 4C).

### 2.5. Effects of WIF on Inflammation-Related mRNA in AD-like Mouse Model

The control group that received DNCB exhibited significantly elevated mRNA levels of inflammatory proteins associated with AD, such as TNF-α, interferon gamma (IFN-γ), interleukin IL-6, and IL-13. Conversely, in mice that received WIF, the mRNA levels of these proteins were significantly suppressed (Figure 5). These results suggest that WIF ameliorates DNCB-induced AD by suppressing the expression of AD-related inflammatory genes and proteins.

### 2.6. Effects of WIF on TNF-α/IFN-γ–Induced Pro-Inflammatory Chemokines Release and mRNA Expression in Human Keratinocytes

In the in vitro cell model of TNF-α/IFN-γ-induced AD in HaCaT cells, the 3-(4,5-dimethylthiazol-2-yl)-2,5-diphenyl-2H-tetrazolium bromide (MTT) assay was performed first to rule out the possibility that the inhibitory efficacy of WIF was due to WIF toxicity in human keratinocyte (HaCaT) cells. At the WIF concentration used (50–200 μg/mL), WIF exhibited no cytotoxicity; instead, cell proliferation slightly increased in a dose-dependent manner (Figure 6A). WIF decreased the protein expression levels of RANTES, TARC, MDC, MCP-1, and MIP-3α in the cell culture media (Figure 6B–F). In addition, WIF inhibited RANTES, MDC, and TARC mRNA expression (Figure 6G–I). These results suggest that WIF exerts an anti-AD effect by inhibiting the generation and expression of RANTES, TARC, MDC, MCP-1, and MIP-3α in an AD-induced in vitro model.

### 2.7. Effect of WIF on TNF-α/IFN-γ-Induced NF-κB p65 Translocation in HaCaT Cells

WIF inhibited p65 translocation from the cytoplasm to the nucleus in the mouse dorsal skin (Figure 4). Based on these results, the anti-inflammatory effects of WIF were revalidated by measuring TNF-α/IFN-γ-induced NF-kB p65 translocation in skin keratinocytes. To measure NF-κB activity, NF-κB p65 subunit translocation from the cytoplasm to the nucleus was monitored. Under basal conditions, the p65 subunit was predominantly distributed in the cytoplasm. However, following TNF-α/IFN-γ stimulation, the p65 subunit underwent significant translocation to the nucleus. The results of TNF-α/IFN-γ stimulation reveal that WIF pretreatment can reduce the nuclear translocation of NF-κB in contrast to TNF-α/IFN-γ treatment (Figure 7).

### 2.8. Identification of Indican and Isovitexin of WIF via High-Performance Liquid Chromatography-Diode Array Detection Analysis

To confirm the contents and identification of two compounds, indican and isovitexin, high-performance liquid chromatography (HPLC) analysis was performed. As shown in Figure 8, prominent peaks in the WIF extract HPLC chromatogram that we analyzed were only two peaks, and these peaks were identified as indican and isovitexin, which compared with the standard compound solution retention time and UV spectrum. Therefore, indican and isovitexin were selected as standard components for analysis. Indican and isovitexin were detected as characteristic peaks that exhibited the highest intensity, minimizing interference from other analyses in WIF. The two compounds were identified via comparison with the retention time (tR; 14.853 and 21.303 min) and ultraviolet (UV) spectrum (280 nm) on the HPLC chromatogram of WIF and the standard component (Figure 8).

### 2.9. Indican and Isovitexin Contents in WIF

To quantify indican and isovitexinin contents in WIF, a standard curve at each concentration was prepared. The standard compound calibration curve exhibited good linearity at the tested concentration range. The area value of indican and isovitexin in WIF was calculated for each standard calibration curve equation. The indican and isovitexin contents were 0.18% and 0.25%, respectively. Indican and isovitexin were detected in the WIF extract, and isovitexin had the highest contents in the WIF extract.

## 3. Discussion

Isatidis folium is known for its antiviral, immunomodulatory, and antipyretic effects against diseases such as influenza, measles, acute infectious hepatitis, dysentery, acute gastroenteritis, and acute pneumonia. Furthermore, it is a well-known herbal medicine that was strongly recommended by the Ministry of Health in China as an effective antiviral prophylactic during the severe acute respiratory syndrome pandemic in 2003 and the H1N1 influenza outbreak in China in 2009. However, studies of the mechanism of anti-AD activity by Isatidis folium leaves are insufficient [22,23]. This study investigated the anti-AD effects of WIF in a DNCB-induced AD-like mouse model, verified the anti-AD effects of WIF in human keratinocyte HaCaT cells, and demonstrated their underlying molecular mechanisms.

The spleen is an important immune organ of the body. Its weight and volume increase with improved function. The spleen plays an important role in AD pathogenesis. Herein, we observed significant increases in spleen weight and size in the DNCB-induced AD-like mouse model, which were alleviated via oral WIF administration. This is partially associated with the immunomodulatory effects of WIF in a DNCB-induced AD-like mouse model. Furthermore, WIF reduced serum IgE and TNF-α levels, which were elevated in the DNCB-induced group, alleviated epidermal and dermal thickness, and suppressed cell migration of inflammatory cells, such as eosinophils and mast cells. Based on these results, it can be considered that WIF exhibited anti-AD efficacy in a DNCB-induced AD-like mouse model [27]. Furthermore, WIF strongly inhibited ERK and p38 MAPKs activities and NF-κB-p65 translocation from the cytoplasm to the nucleus in the dorsal skin tissues of a DNCB-induced AD-like mouse model.

NF-κB is a transcription factor closely associated with cytokine-mediated inflammatory responses and participates in inducing the expression of various pro-inflammatory genes, including those encoding cytokines and chemokines [28]. MAPK signaling also plays an important role in inflammatory cytokine and chemokine production. The findings of this study indicate that WIF effectively reduced inflammatory and immune responses by inhibiting the activation of MAPK and NF-κB signaling pathways, further suggesting that WIF alleviates AD by regulating multiple signaling pathways and downregulating chemokines and chemokine production. In addition, damage to the skin barrier increases the permeability of natural allergens and artificial pollutants, consequently exacerbating the AD symptoms [3,9]. Although the precise mechanisms underlying AD pathogenesis remain unclear, the complex interplay between inflammation, barrier dysfunction, and pruritus is important in its development, progression, and chronicity. MIP-3α is a C-C chemokine predominantly expressed in extra lymphatic tissues and reportedly directs the migration of dendritic cell precursors and memory lymphocytes to antigen invasion sites [29]. It is also a chemokine upregulated in the skin in cases of disrupted skin barrier and inflammatory skin conditions with an impaired barrier function. It is further associated with the weakening of the skin barrier against infections and AD [13,30]. WIF inhibits MIP-3α production, which was hypothesized to affect the skin barrier.

As in other reported papers, indican and isovitexin were detected in our WIF extract too, and the contents were similar [16,17,26]. It was suggested that indican and isovitexin are major components of WIF. However, the main compound may not be the active compound, and we have not yet investigated the effects of indican and isovitexin on AD. Therefore, further studies are needed to identify active compounds in the AD efficacy of WIF.

## 4. Materials and Methods

### 4.1. Preparation of WIF

Isatidis folium were purchased from Yeong-cheon Oriental Herbal Market (Yeongcheon, Republic of Korea). Hot water extraction and drug preparation methods were the same as in a previous study [31]. For in vivo experiments, WIF powder was used, dissolved in 0.5% carboxymethyl cellulose (CMC) solution.

### 4.2. HPLC Instrument and Condition

Indican (Indoxyl β-D-glucoside) and isovitexin were purchased from Sigma-Aldrich (St. Louis, MO, USA). The organic solvent used in HPLC analysis, acetonitrile, was purchased from Merck KgaA (Darmstadt, Germany). ACS reagent grade formic acid was obtained from Sigma-Aldrich (St. Louis, MO, USA). Indican and isovitexin were dissolved in methanol at 1.0 mg/mL and diluted to obtain different concentrations for preparing the standard curve. WIF was precisely weighed and dissolved using water (HPLC grade) at a concentration of 10 mg/mL. All prepared solvents were filtered through a 0.2 μm membrane filter. HPLC analysis was performed using the Dionex UltiMate 3000 system (Dionex Corp., Sunnyvale, CA, USA) equipped with a binary pump, autosampler, column oven, and diode array UV/VIS detector (DAD). System control and data analysis were processed using a Dionex Chromeleon. The analysis conditions were as follows: X bridge C18 column (250 × 4.60 mm, 5 μm, Waters, Milford, MA, USA), the column oven temperature was maintained at 25 °C. The mobile phase consisted of 0.1% (*v*/*v*) formic acid in water (A) and acetonitrile (B) gradient elution system to improve the chromatographic separation capacity. The gradient system was programmed as 5% B, 0–2 min; 5–40% B, 2–40 min; and 40–100% B, 40–50 min, at a 1.0 mL/min flow rate. The detection wavelength was 280 nm, and all the samples were injected under the same conditions and triplicated.

### 4.3. AD-Like Mouse Model and Histological Analysis

Mice used in this experiment were 6-week-old male BALB/c mice (about 18–20 g) purchased from Samtako Bio Korea (Osan, Republic of Korea). All animal experiments were approved by the Institutional Animal Care and Use Committee (IACUC; D-21-044) of the Korea Institute of Oriental Medicine (KIOM). All animals were maintained under constant conditions of 42.6 ± 1.7% humidity, 22.5 ± 0.5 °C temperature, and a 12 h light-dark cycle. The mice were randomly divided into five groups: normal (negative control), control (positive control; DNCB sensitized), WIF 100 mpk (DNCB + 100 mg/kg WIF), WIF 200 mpk (DNCB + 200 mg/kg WIF), and DEX (dexamethasone; DNCB + 1 mg/kg DEX; positive control). For anesthesia, about 250 mg/kg of avertine (2,2,2-tribromoethanol) diluted in 2-methyl-2-butanol was intraperitoneally injected. Following euthanization, the dorsal skin tissues of the mice were collected and fixed in 10% neutral buffered formalin at room temperature for 24 h and embedded in paraffin. The fixed samples were submitted to a commercial company (Garam Meditech, Dae-gu, Republic of Korea) for H&E and toluidine blue staining. Mast cells and eosinophils infiltrating the dermis were observed under light microscopy, and the skin thickness and number of inflammatory cells were measured using ImageJ version 1.46 (National Institutes of Health). The experimental method was performed in the same way as previously reported [32].

### 4.4. Cell Culture and Cytotoxicity Assay

HaCaT cells were carefully cultured in Dulbecco’s modified Eagle’s medium comprising 10% (*v*/*v*) fetal bovine serum and antibiotics (100 U/mL penicillin and 100 μg/mL streptomycin) at 37 °C in a humidified 5% CO_2_ incubator. Cells from the third passage were used for the experiments. Cell viability was evaluated using EZ-Cytox assay (Daeil Lab, Seoul, Republic of Korea). HaCaT cells (8 × 10^3^ cells/well) were seeded in 96-well plates, stabilized for 24 h, replaced with a serum-free medium containing various WIF concentrations (50, 100, and 200 μg/mL), and incubated for 24 h. Then, EZ-Cytox solution was added and incubated for an additional 2 h. Following incubation, absorbance was determined at 450 nm using a spectrophotometer (Infinite M200 Tecan, Männedorf, Switzerland).

### 4.5. Biochemical Parameters

Serum IgE and TNF-α (BD, East Rutherford, NJ, USA) levels were determined using commercial ELISA kits according to the manufacturers’ protocols. In addition, the cell supernatant samples were obtained from HaCaT cells. The cells (8 × 10^4^ cells/well) were seeded into 24-well plates, maintained at 37 °C in 5% CO_2_, and stabilized for 24 h for cell attachment. Following stabilization, the cells were pretreated with 50, 100, and 200 μg/mL WIF for 1 h, stimulated with 10 ng/mL TNF-α/IFN-γ (R&D Systems, Minneapolis, MN, USA), and incubated for 24 h. Thereafter, the level of chemokine production, such as RANTES and TARC (Bio-Legend ELISA kit, San Diego, CA, USA), MCP-1 (BD, USA), MDC and MIP-3α (R&D Systems, USA), was estimated using the culture media. All the experiments were performed at least in triplicate according to the manufacturer’s protocol.

### 4.6. Real-Time RT-PCR Analysis

Total RNA was extracted from dorsal skin samples and HaCaT cells using TRIzol reagent (Invitrogen Life Technologies, Carlsbad, CA, USA) according to the manufacturer’s protocol. For cell experiments, WIF was pretreated at 50, 100, and 200 μg/mL for 1 h (4 × 10^5^ cells/well) and stimulated using TNF-α/IFN-γ for 24 h. After washing and harvesting the cells, the total RNA was extracted, and cDNA was synthesized. Real-time RT-PCR was performed using the synthesized cDNA and specific primers for the target genes. RT-PCR and real-time RT-PCR were performed according to the manufacturer’s instructions (Bioneer, Daejeon, Republic of Korea). cDNA was synthesized using 1 μg total RNA as a template. PCR conditions were as follows: 40 cycles of initial denaturation at 95 °C for 5 s, followed by primer annealing at 62.5 °C for 30 s. The target genes were quantified via the 2^−ΔΔCT^ method. The primer sequences were indicated in previous studies [32].

### 4.7. Western Blot Analysis

A portion of mouse dorsal skin tissue was lysed using NE-PER™ Nuclear and Cytoplasmic Extraction Reagent for nuclear fraction extraction. In addition, T-PER™ Tissue Protein Extraction Reagent and Extraction Buffer was used for total protein extraction for another portion of the dorsal skin tissue. Proteins were obtained via the centrifugation of the lysate at 12,000 rpm at 4 °C for 10 min. Proteins (30 μg) were separated via sodium dodecyl sulfate-polyacrylamide gel electrophoresis and electrophoretically transferred onto a nitrocellulose (NC) membrane (Whatman, Maidstone, UK). Protein-transferred NC membranes were blocked with 5% skim milk or 3% bovine serum albumin (BSA; GenDEPOT, USA) for 1 h at room temperature. Membranes were prepared with primary antibodies, such as p-ERK/t-ERK (p-ERK #4370, t-ERK #9102; Cell Signaling Technology, Danvers, MA, USA), p-p38/t-p38 (p-p38 #9211, t-p38 #9212; Cell Signaling Technology, USA) and NF-κB/Lamin B1 (NF-κB #8242, Lamin B1 #12586; Cell Signaling Technology, USA), diluted at 1:1000 in 3% BSA, and incubated overnight at 4 °C, followed by secondary antibodies incubated for 1 h at room temperature. The expression of specific proteins was detected as bands via Pierce™ ECL Western Blotting Substrate (Thermo Fisher Scientific, Waltham, MA, USA) and quantified using ImageJ (version 1.51j8, National Institutes of Health).

### 4.8. Immunocytochemistry Analysis

HaCaT cells were seeded in confocal (glass-bottom) dishes and stabilized for 24 h for cell attachment. Following incubation for 24 h, the cells were treated with WIF (50, 100, and 200 µg/mL) for 1 h and stimulated using TNF-α/IFN-γ (10 ng/mL each) for 20 min. The stimulated HaCaT cells were fixed with 4% paraformaldehyde and incubated overnight with NF-κB p65 antibodies (Cell Signaling Technology, USA) at 4 °C, followed by the addition of Alexa Fluor 488 secondary antibody (Thermo Fisher Scientific, USA) and incubation for 20 min at room temperature. The nuclei were stained with DAPI solution (Sigma-Aldrich, USA) at room temperature for 10 min. The cells were visualized under a confocal microscope (FV3000 FLUOVIEW; Olympus, Tokyo, Japan).

### 4.9. Statistical Analysis

All experiments were repeated at least in triplicate. Statistical analysis was performed using Prism version 5.01 (GraphPad Software, Inc., San Diego, CA, USA). The results were presented as mean ± standard error of the mean assessed using Student’s *t*-test or analysis of variance (Dunnett’s, control comparison). A *p*-value of 0.05 was considered statistically significant.

## 5. Conclusions

Herein, in a DNCB-induced AD-like mouse model, WIF improved AD symptoms by reducing immune cell infiltration and suppressing the generation and expression of inflammation-related factors. Furthermore, WIF inhibited inflammation-related mediators by inhibiting the ERK, p38 MAPK, and NF-κB signaling pathways. These findings suggested the potential of WIF as an effective AD treatment.

## Figures and Tables

**Figure 1 molecules-28-03960-f001:**
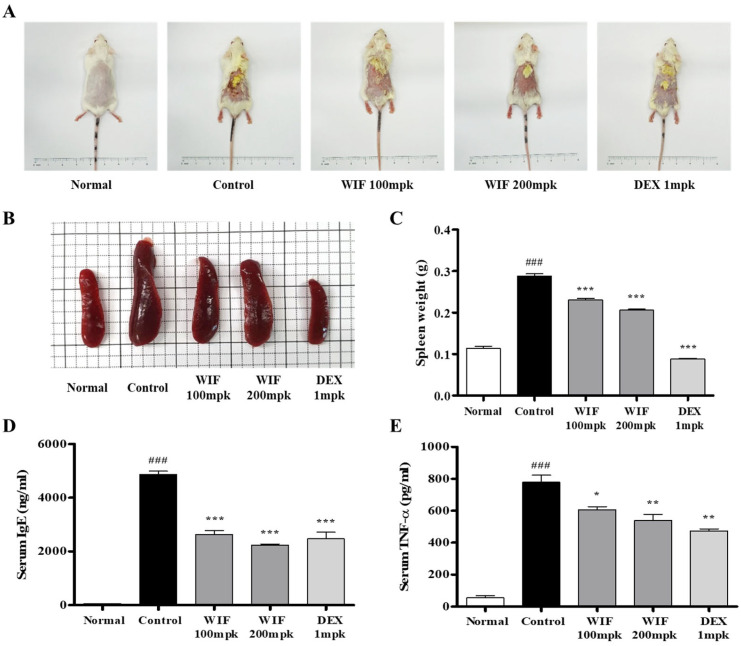
Effects of WIF on AD-like symptoms in DNCB-induced Balb/c mice. (**A**) Representative clinical features. (**B**,**C**) Spleen weights. (**D**,**E**) Serum IgE and TNF-α levels were determined via enzyme-linked immunosorbent assay (ELISA). Results are presented as mean ± standard deviation (SD; *n* = 10 per experiment). ^###^
*p* < 0.001 vs. normal group; * *p* < 0.1, ** *p* < 0.01, *** *p* < 0.001 vs. DNCB-induced group. mpk, mg/kg.

**Figure 2 molecules-28-03960-f002:**
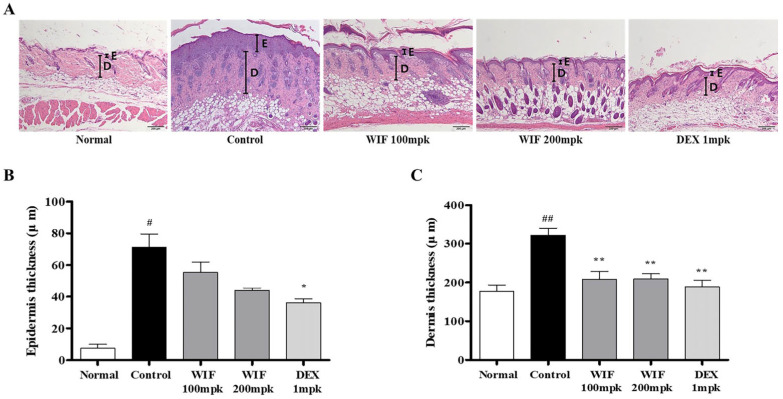
Effects of WIF on epidermis and dermis thickness in DNCB-induced AD skin lesions. (**A**) Histopathological features of the dorsal skin via H&E staining (100× magnification; scale bar: 200 µm). (**B**,**C**) Determination of epidermal and dermal thickness. Results are presented as mean ± SD (*n* = 10 per experiment). ^#^
*p* < 0.05, ^##^
*p* < 0.01 vs. normal group; * *p* < 0.05, ** *p* < 0.01, vs. DNCB-induced group. E, Epidermis; D, Dermis.

**Figure 3 molecules-28-03960-f003:**
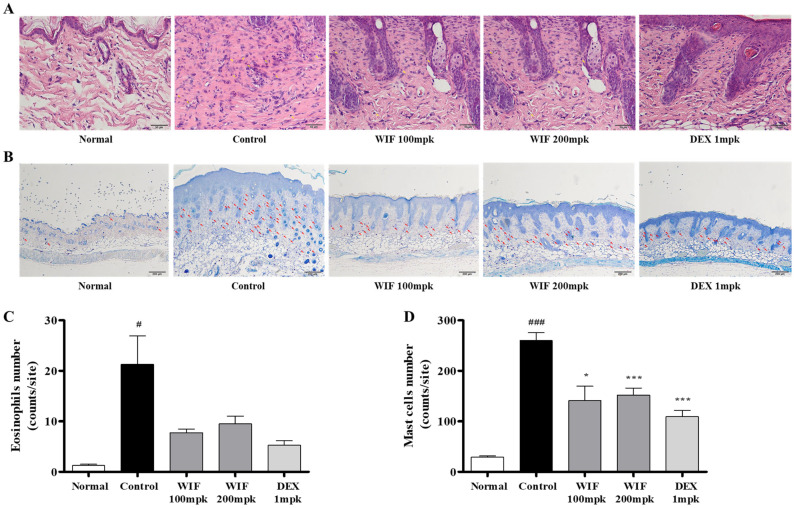
Effects of WIF on histological features in DNCB-induced AD skin lesions. (**A**) Eosinophil infiltration was confirmed via H&E staining (magnification, ×400; scale bar, 50 µm). (**B**) Mast cell infiltration was confirmed via toluidine blue staining (magnification, ×100; scale bar, 200 µm). (**C**,**D**) Determination of eosinophil and mast cell infiltration. Results are presented as mean ± SD (*n* = 10 per experiment). ^#^
*p* < 0.05, ^###^
*p* < 0.001 vs. normal group; * *p* < 0.05, *** *p* < 0.001 vs. DNCB-induced group.

**Figure 4 molecules-28-03960-f004:**
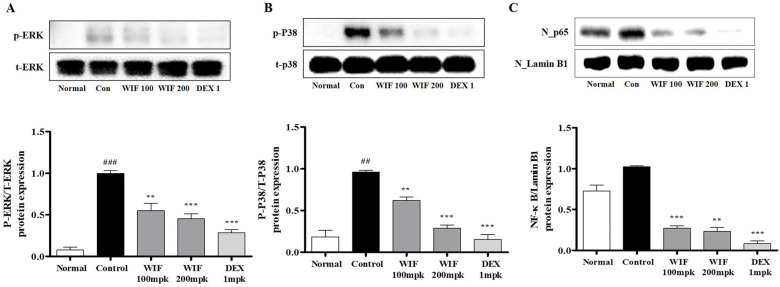
Effects of WIF on MAPK/NF-κB activation in DNCB-induced AD skin lesions. (**A**–**C**) Expression of MAPK (ERK and p38) and NF-κB was determined via Western blot analysis. Results are presented as mean ± SD (*n* = 5 per experiment). ^##^
*p* < 0.05, ^###^
*p* < 0.001 vs. normal group; ** *p* < 0.01, *** *p* < 0.001 vs. DNCB-induced group.

**Figure 5 molecules-28-03960-f005:**
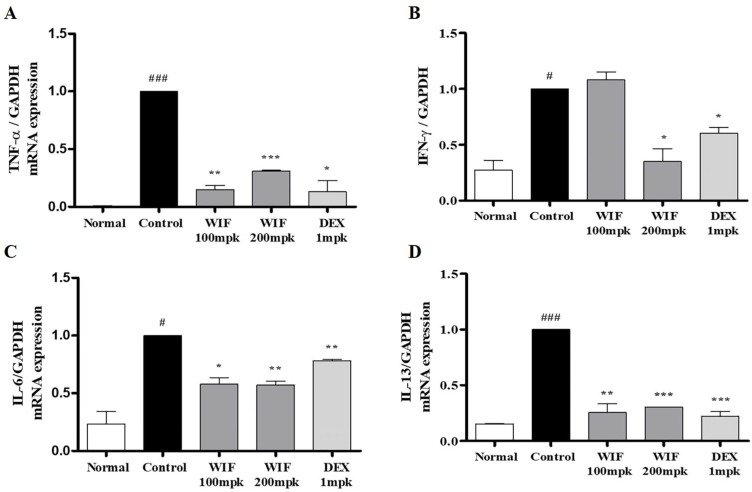
Effects of WIF on AD-induced cytokine activation in DNCB-induced AD-like skin lesions. (**A**–**D**) Levels of TNF-α, IFN-γ, IL-6, and IL-13 production were determined via real-time reverse transcription-polymerase chain reaction (RT-PCR). Results are presented as mean ± SD (*n* = 5 per experiment). ^#^
*p* < 0.01, ^###^
*p* < 0.001 vs. normal group; * *p* < 0.05, ** *p* < 0.01, *** *p* < 0.001 vs. DNCB-induced group.

**Figure 6 molecules-28-03960-f006:**
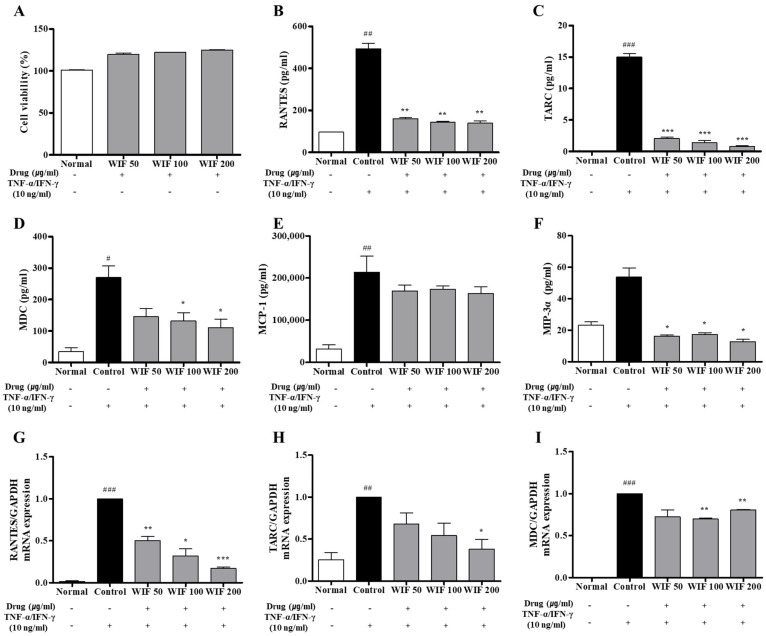
Effects of WIF on chemokine levels in TI-stimulated HaCaT cells. (**A**) WIF cytotoxicity was determined via the EZ-Cytox assay. (**B**–**F**) Levels of RANTES, TARC, MDC, MCP-1, and MIP-3α levels were examined via ELISA. (**G**–**I**) Production levels of RANTES, TARC, and MDC were determined via real-time RT-PCR. Results are presented as mean ± SD (*n* = 3 per experiment). ^#^
*p* < 0.1, ^##^
*p* < 0.01, ^###^
*p* < 0.001 vs. normal group; * *p* < 0.05, ** *p* < 0.01, *** *p* < 0.001 vs. TI-stimulated group.

**Figure 7 molecules-28-03960-f007:**
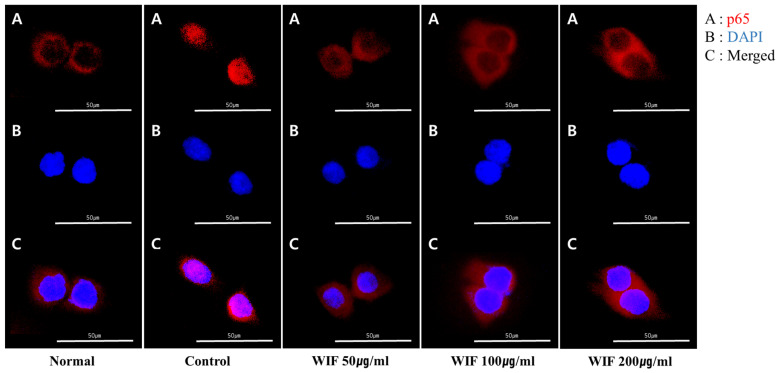
Effects of WIF on NF-κB p65 translocation in TI-stimulated HaCaT cells. (**A**–**C**) Immunofluorescence staining of p65 (red) was analyzed using a fluorescent microscopy. 4′,6-Diamidino-2-phenylindole (DAPI; blue) staining was conducted to determine the location of the nucleus. Scale: 50 µm.

**Figure 8 molecules-28-03960-f008:**
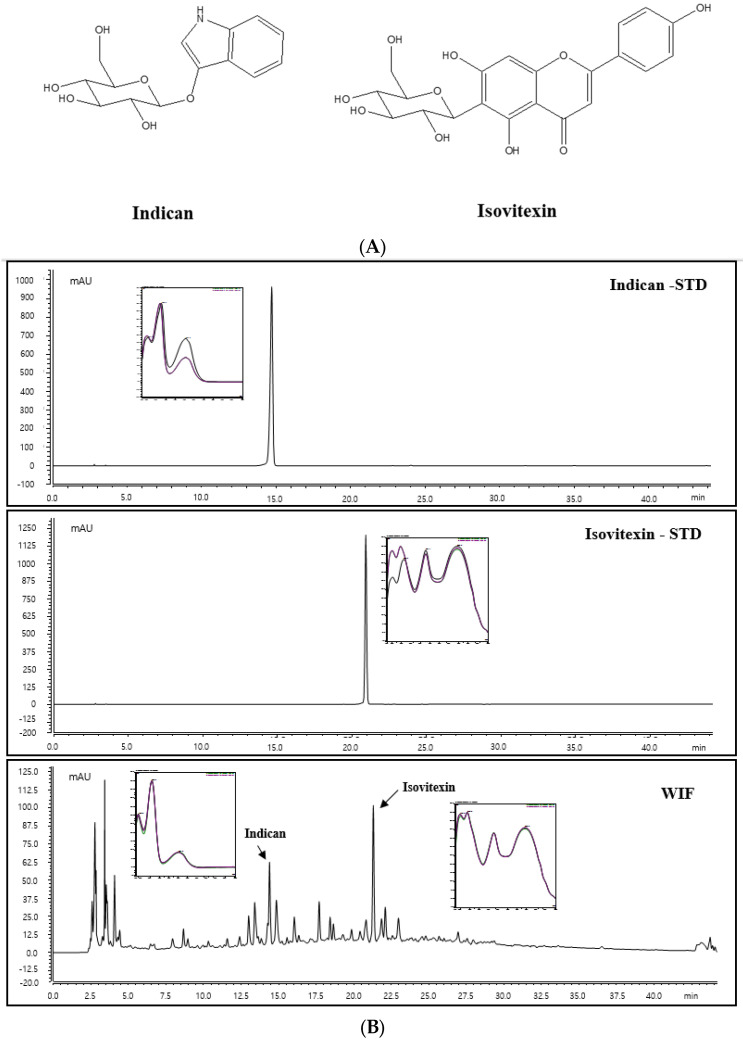
(**A**) Chemical structures of indican and isovitexin. (**B**) Standard indican and isovitexin solutions and the WIF extract.

## Data Availability

The data presented in this study are available on request from the corresponding author.

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
