# Peer review of "Anti-Atopic Effect of Isatidis Folium Water Extract in TNF-α/IFN-γ-Induced HaCaT Cells and DNCB-Induced Atopic Dermatitis Mouse Model"

_molecules, 2023, doi:10.3390/molecules28093960_

Round 1

Reviewer 1 Report

The Authors examined potential of Isatidis folium to ameliorate symptoms of atopic dermatitis. As a prevalence of this disease is increasing and AD pose a significant burden on health-care resources and quality of life of patients, there is a need to find a more effective and lower-cost therapy. The manuscript is interesting, but I have several comments.

1. Abstract should be rewritten. For example lines 16-18 are not clear “After 30 days of induction and oral administration…” what was orally administered? Lines 18-23: to my opinion there are lot methodological data (confirmed by: H&E and Toluidine Blue staining, ELISA kit and PCR, by western blot etc.) that can be shortened. Additionally, it is not clear what type of changes was noted (increase or decrease).

2. Please change “vehicle” to “normal” on all graphs regarding mouse model. DNCB was dissolved in acetone/olive oil thus water is not appropriate vehicle. As acetone cause epithelial barrier disruption and can impact cytokine production (for example IL-6), my suggestion is to remove significance between vehicle and control (DNCB treatment) and to present differences between dexamethasone and WIF treatments. This statistic can indicate whether WIF is more effective than standard therapy.   

3. Please check significance between control and treatment. In Fig 3D it seems that treatment lower number of infiltrating eosinophils.

4. Page 5, lines 155-158. Two similar sentences. Please delete one.

5. Did the Authors measured IL-17 in skin. This cytokine is elevated in various skin disorders including atopic dermatitis.

6. What is rationale for determination of indican and isovitexin in WIF extract? Are there other compounds in extract? Did the Authors examine effect of indican and isovitexin on AD? To my opinion, this paragraph should be deleted.

7. Discussion should be rewritten as large portion of this segment is more appropriate for Introduction. Additionally, there is no need to recapitulate results (page 10, lines 248-249).

8. Please provide manufacturers data (lines: 321 (TNF/IFN-γ), , 323-324 (ELISA kits for chemokines), 347-349 (antibodies for western blot).

9. Spell check and grammar check are needed.

Author Response

We would like to thank an editor for your kind comments and for the thorough review of this paper. Also, we were attempt to address in your respective concerns in the following the same order as your discussion.

Please see the attachment file.

Reviewer 2 Report

A nice piece of work for which I commend the authors. For the sake of clarity and to improve the quality of the manuscript, I suggest the following corrections for the authors' consideration. L prefix refers to the Line number of the manuscript.

Isatis tinctoria and I. tinctora should be italicised at all instances.

Line 59: ‘particulates’ is better replaced with ‘factors’

L84: Explain what MMP stands for.

2.8. The content of two constituents in WIF water extract: This para (L198 – 215) is very poorly written (especially grammar) and is difficult to comprehend. It needs re-writing for clarity.

L273: “ACS reagent-grade formic acid was obtained Sigma Aldrich (St. Louis, MO, USA).” Should be corrected as “ACS reagent-grade formic acid was obtained from Sigma Aldrich (St. Louis, MO, USA).”

L274-75 “…and isovitexin was dissolved methanol at 1.0mg/mL and diluted at each concentration for prepared standard curve.” Should be corrected as “…and isovitexin were dissolved in methanol at 1.0mg/mL and diluted to obtain different concentrations for preparing the standard curve.”

L280: “The analysis conditions were followed by:….” Should be corrected as “The analysis conditions were as follows:…”

L285-86: “A detection wavelength was 280nm and all samples were injected under same condition and triplicated. Should be corrected as The detection wavelength was 280nm and all samples were injected under the same condition, in triplicate.

L290: Instructional Animal Care and Use Committee”. Is this correct or should this be  “Institutional Animal Care and Use Committee”?

L299: “commissioned” to be corrected as “submitted”

L304: “…previous method” to be corrected as “…previously published”

L312: “After then,” change to “Then,”

L327: “Total RNA from dorsal skin samples…” change to “Total RNA was extracted from dorsal skin samples….”

L342: “a part of the other dorsal skin tissue.” replace with “another portion of the dorsal skin tissue.”

L344: “…electrophoretic transferred…” should be “electrophoretically transferred…”

L354: “pre-treated with WIF..” should this be treated with WIF…”?

L355-59: “Stimulated HaCaT cells fixed with 4% paraformaldehyde and incubated overnight with NF-κB p65 antibodies (Cell Signaling Technology, Inc., USA) at 4 °C, followed by Alexa-Fluor-488 secondary antibody (Thermo Fisher Scientific Inc., USA) were incubated for 20 min at room temperature.” Should be corrected as “Stimulated HaCaT cells were fixed with 4% paraformaldehyde and incubated overnight with NF-κB p65 antibodies (Cell Signaling Technology, Inc., USA) at 4 °C, followed by the addition of Alexa-Fluor-488 secondary antibody (Thermo Fisher Scientific Inc., USA) and incubation for 20 min at room temperature.”

L391: Informed Consent Statement: “Informed consent was obtained from all subjects involved in the study.” As the research was done with mice and not human subjects, should the response be “Not applicable”?

Author Response

We would like to thank an editor for your kind comments and for the thorough review of this paper. Also, we will attempt to address in your respective concerns in the following the same order as your discussion.

Please see the attachment file.

Round 2

Reviewer 1 Report

I have no additional comments

Author Response

We would like to express my sincere thanks to you for nice comments and suggestions. We addressed all comments and revised the manuscripts. You can find on page 2, lines 205-09 and on page 10, lines 267-72.
